# Applied Methods to Assess the Antimicrobial Activity of Metallic-Based Nanoparticles

**DOI:** 10.3390/bioengineering10111259

**Published:** 2023-10-28

**Authors:** Etelka Chung, Guogang Ren, Ian Johnston, Rupy Kaur Matharu, Lena Ciric, Agnieszka Walecka, Yuen-Ki Cheong

**Affiliations:** 1Centre for Engineering Research, University of Hertfordshire, Hatfield AL10 9AB, UK; e.chung@herts.ac.uk (E.C.); i.d.johnston@herts.ac.uk (I.J.); 2Department of Mechanical Engineering, University College London, Torrington Place, London WC1E 7JE, UK; rupy.matharu.15@ucl.ac.uk; 3Department of Civil, Environmental and Geomatic Engineering, University College London, Gower Street, London WC1E 6BT, UK; l.ciric@ucl.ac.uk; 4Intensive Care Unit, Royal Free Hospital, Royal Free London NHS Foundation Trust, Pond Street, London NW3 2QG, UK; agnieszka.walecka@nhs.net

**Keywords:** antimicrobial nanoparticles, minimum inhibitory concentration (MIC), live–dead assay, resazurin, silver, copper

## Abstract

With the rise of antibiotic resistance, the drive to discover novel antimicrobial substances and standard testing methods with the aim of controlling transmissive diseases are substantially high. In healthcare sectors and industries, although methods for testing antibiotics and other aqueous-based reagents are well established, methods for testing nanomaterials, non-polar and other particle-based suspensions are still debatable. Hence, utilities of ISO standard validations of such substances have been recalled where corrective actions had to be taken. This paper reports a serial analysis obtained from testing the antimicrobial activities of 10 metallic-based nanomaterials against 10 different pathogens using five different in vitro assays, where the technique, limitation and robustness of each method were evaluated. To confirm antimicrobial activities of metallic-based nanomaterial suspensions, it was found that at least two methods must be used, one being the agar well diffusion method, which was found to be the most reliable method. The agar well diffusion method provided not only information on antimicrobial efficacy through the size of the inhibitory zones, but it also identified antimicrobial ions and synergistic effects released by the test materials. To ascertain the effective inhibitory concentration of nanoparticles, the resazurin broth dilution method is recommended, as MIC can be determined visually without utilising any equipment. This method also overcomes the limit of detection (LoD) and absorbance interference issues, which are often found in the overexpression of cell debris and nanoparticles or quantum dots with optical profiles. In this study, bimetallic AgCu was found to be the most effective antimicrobial nanoparticle tested against across the bacterial (MIC 7 µg/mL) and fungal (MIC 62.5 µg/mL) species.

## 1. Background

Materials with antimicrobial activity have the ability to inhibit or destroy the microbes that are in contact or within a certain distance [1]. Commonly, antimicrobial compounds and solutions are used to disinfect surfaces, equipment and devices from pathogens, whilst antibiotics are used to treat infections. In healthcare settings, the decontamination process is particularly important as it can help to lower infection rates. With the rise of antimicrobial resistance, resulting in ineffective antibiotics, procedures to assist in the control of infectious pathogens are in high demand. Therefore, the assessment of antimicrobial activity is important to determine the effectiveness of the materials prior to their use to help us to control and effectively reduce the spread of microbes [2,3].

Currently, there are numerous methods used to analyse the antimicrobial effects of compounds, with certain standards recognised by the Clinical Laboratory Standards Institute (CLSI) and the European Committee on Antimicrobial Susceptibility Testing (EUCAST) [2]. Each analytical method has its own advantages and disadvantages; however, it is difficult to compare the antimicrobial quantification between different methods, and limitations have been shown when tested with nanoparticle suspensions [2,4].

Nanoparticles, with structures typically less than 100 nm, have been investigated for their antimicrobial activity against a range of microbes, especially for their ability to inhibit certain microbes with developed antimicrobial resistance [5]. In addition, many nanoparticle compositions, such as metal- and carbon-based, have shown no toxicity toward mammalian cells, giving great potential for a range of biomedical applications [6,7]. Currently, silver nanoparticles are being used in medical applications for their antimicrobial properties; such applications include dental implants, device coatings and wound dressing [8]. Whilst there is no standard protocol to investigate the antimicrobial properties of nanoparticles, a variety of methods have been used in published papers. For example, to investigate minimal inhibitory concentration (MIC), Loo et al. (2018) used resazurin solution to indicate bacterial growth [9]. In contrast, Ruparelia et al. (2008) used a spectrometer to measure bacterial growth through increased absorbance to determine MIC [10]. Although both reports tested Ag nanoparticles with diameters of 4.06 nm and 3 nm, respectively, against *E. coli*, the MIC varied from 7.8 µg/mL to 180 µg/mL, respectively. Therefore, it is necessary to investigate different antimicrobial testing methods for nanoparticles and to evaluate the level of precisions in each analysis.

Herein, ten nanoparticles, including monometallic, bimetallic, intermetallic and metal oxides, were tested against ten of the most common nosocomial microbial strains to assess their antimicrobial activity. These nanoparticles were selected due to evidence in the literature of their antimicrobial activity against several species of microbes [11,12,13,14]. Five different antimicrobial assays were used and compared to identify nanoparticles with antimicrobial activities. The agar well diffusion method identifies antimicrobial activity through visible rings of inhibition, whilst colorimetric approaches that use resazurin in the broth and microtitre assays provide robust screening and MIC results. Finally, the microbial growth rates and viabilities were analysed using optical spectroscopy and Fluorescent Live/Dead techniques, respectively.

## 2. Methods and Materials

### 2.1. General Materials and nanoparticle Preparations

The nanoparticles were manufactured or engineered by various suppliers: Cu10, Cu60 and CuO40 were purchased from Canfuo Nano Technology ^®^ (Suzhou, China); Ag100, Cu90 and CuO10 were engineered by Intrinsiq Materials ^®^ (Farmborough, UK) using Tesima™ Plasma process [13]; Ag20 and ZnO were purchased from Alfa Aesar (Heysham, UK); and lastly, AgCu and CuZn were purchased from Sigma-Aldrich (Dorset, UK). The number following the nanoparticle name corresponds to the average particle diameter size in nm. All nanoparticles are spherical in morphology, except Cu10, which is cubical and CuO40, which is rod-shaped. The morphology, chemical compositions and antiviral evaluations of the bimetallic AgCu and intermetallic CuZn were previously reported [15]. The morphology, chemical properties and antimicrobial and antiviral activity of Ag100 and Cu90 were previously reported [16].

Nanoparticle stock dispersions were prepared by measuring the desired mass of the nanoparticles and sonication using a high-frequency liquid processor (Sonics & Materials^®^, Masssachusetts (MA), USA) with corresponding volumes of extra pure deionized water (Acros Organics, Loughborough, UK) and then immediately cooled using cold tap water.

### 2.2. Microbial Cultures

Microorganisms *Acinterbacter baumannii* (ATCC 19606), *Pseudomonas aeruginosa* (ATCC 25668), *Klebsiella pneumonia* (ATCC 15380), *Escherichia coli*, *Enterococcus faecalis* (ATCC 29212), *Staphylococcus aureus*, *Salmonella typhimurium*, *Streptococcus pyogenes*, *Candida albicans* (ATCC 2091), *C. tropicalis* (ATCC 20336) were obtained from The University of Hertfordshire microorganism collection. Bacterial and fungal stock cultures were grown on nutrient and yeast peptone dextrose agar plates, respectively, and then grown and diluted with corresponding broths for desired dilutions.

### 2.3. Agar Well Diffusion

Microbial cell numbers were adjusted to ~3 × 10^7^ CFU/mL using spectrometry absorbance at 600 nm (0.1 OD for bacterial cells and 1.0 OD for *Candida* spp.) and spread onto Mueller–Hinton agar (Sigma-Aldrich, Dorset, UK). Wells were made using a corker, and nanoparticle suspensions at 1000 µg/mL were pipetted into the wells. The agar plates were then incubated at 37 °C for 24 h. The zone of inhibition was observed, and the diameters were measured in cm. Three replicates were conducted for each sample and mean results were calculated. A representative sample of an agar well diffusion plate experiment and further detail on the zone of inhibition measurement can be found in the Appendix A.

### 2.4. Resazurin Broth Assay

Microbial cells were adjusted to ~3 × 10^7^ CFU/mL and added to a 96-well plate. Nanoparticle suspensions were added to the plate to make up a final concentration of 1000 µg/mL. The plates were incubated at 37 °C for 24 h. Resazurin dye (0.02%) (Sigma-Aldrich, Dorset, UK) was added to wells and incubated for a further 24 h at 37 °C. The viability of cells was assessed through colour change of resazurin from blue/purple to fluorescent pink. Three replicates were conducted for each sample.

### 2.5. Resazurin Microtitre MIC Assay

Selected nanoparticle suspensions were added to the top row of a 96-well plate and 2-fold diluted down to form a final concentration gradient from 250 µg/mL to 2.0 µg/mL. Microbes were adjusted to ~10^4^ CFU/mL and added to wells before incubating at 37 °C for 24 h. Resazurin dye (0.02%) was added to wells and incubated for a further 24 h at 37 °C. The lowest concentration of nanoparticles with no change in colour of resazurin was determined as the MIC. Three replicates were conducted for each sample concentration.

### 2.6. Spectrometer Growth Rate

AgCu nanoparticle suspensions were added to the top row of a 96-well plate and diluted 2-fold down to make up a final concentration of 64 µg/mL to 0.5 µg/mL. Microbial cell numbers were adjusted to ~10^4^ CFU/mL and added to wells. A spectrophotometer (BMG CLARIOstar, Aylesbury, UK) was used to monitor the microbial growth by measuring the absorbance (abs) at 600 nm every hour with a 100 rpm orbital shake prior to every measurement and continuous incubation at 37 °C. Initial absorbance was used as a blank. The absorbance of nanoparticles at different concentrations has been considered during data analysis; further information can be found in Appendix A. Four replicates were conducted for each sample concentration, and mean absorbance was presented.

### 2.7. Fluorescent Cell Viability

Microbial cell numbers were adjusted to ~10^7^ CFU/mL and incubated with AgCu nanoparticle suspensions at a concentration of 100 ug/mL and the corresponding MIC value. At every hour, 90 µL of the cultures was taken and treated on ice with 5 µL of propidium iodide (400 µM) and 5 µL of SYTO9 (33.4 µM) dye (Thermo Fisher Scientific, Loughborough, UK) for 15 min with agitation at 100 rpm. Then, 10 µL of treated samples were transferred onto a glass slide and viewed using a fluorescent confocal microscope with corresponding filters (Nikon, New York, NY, USA). The number of live (fluorescent green) and dead (fluorescent red) cells were counted using ImageJ v1.53 (National Institution of Health, Bethesda, MD, USA). Three areas of the slide of two samples were counted, and the results were presented as the mean percentage of live cells and with error bars denoting ½ standard deviation.

## 3. Results

Two robust methods, agar well diffusion and resazurin broth assay, were used to preliminarily evaluate the antimicrobial activities of 10 different nanoparticle suspensions against 10 microbes. To further evaluate antimicrobial activity, the MIC assay was performed on the top five of the highest-performing nanoparticles based on the previous methods. As the bimetallic AgCu nanoparticles were found to have the highest antimicrobial efficacy, further antimicrobial investigations of these nanoparticles were performed using the spectrometer growth rate and fluorescent live–dead assay methods.

### 3.1. Evaluation of Zone Inhibitory Effect on Solid Agar

As shown in Table 1, the diameter of the zone of inhibition was measured in cm after a 24 h incubation with nanoparticles in the cut wells of the agar. Nanoparticles containing silver, including Ag20, Ag100 and AgCu, had a greater antimicrobial effect in comparison to other tested nanoparticles. It is worth noting that the disk diffusion method was initially used as it is one of the most common methods for screening antibiotics. However, no antimicrobial activity was observed when applying the nanoparticle suspensions.

Typically, copper-based nanoparticles (i.e., Cu10, Cu60, Cu90, CuO10 and CuO40) have the largest zone of inhibition, but they were only effective against *P. aeruginosa.* At 1000 µg/mL, the bimetallic AgCu nanoparticle suspension was the only nanoparticle to show antimicrobial activity against all 10 of the tested microbes, with the highest activity against *S. pyogenes*. This showed that out of the tested nanoparticles, bimetallic AgCu nanoparticle suspensions had the broadest antimicrobial activity.

### 3.2. Resazurin Broth Assay

The resazurin assay was used to screen cell viability after the microbes were exposed to 1000 µg/mL of the nanoparticle suspensions for 24 h in broth. Colour change was observed and recorded in Table 2. A visible colour change from blue to pink indicated active, viable cells that caused the reduction of the resazurin to give resorufin, whilst samples that remained blue after incubation indicated no viable cells [17,18]. Cu10 and AgCu nanoparticles showed antimicrobial results toward all of the tested microbes, whilst ZnO had the least antimicrobial effect, with activity toward 30% of the tested microbes. The results of the remaining nanoparticles varied from 90 to 40% effectiveness toward the tested microbes.

### 3.3. Resazurin Microtitre MIC Plate Assay

Based on the results from the agar well diffusion and resazurin broth assays, five of the highest-performing antimicrobial nanoparticles were selected for further testing to determine the MIC. As shown in Table 3, Ag-containing nanoparticle suspensions were more effective (lower MIC values) than Cu-based nanoparticle suspensions. An example of the well plate with results can be found in the Appendix A. Cu-based nanoparticles all had similar efficacy, with typical MIC values ranging from 31 to 250 µg/mL. On the other hand, Ag and AgCu nanoparticles had lower MIC values, which ranged between 7 and 62.5 µg/mL, except for Ag nanoparticles against *E. coli*, with a value of 250 µg/mL.

### 3.4. Spectrometer Growth Rate

The resulting screening and MIC results demonstrated that bimetallic AgCu nanoparticles had the highest and most generic antimicrobial activities and were thus selected for subsequent investigations. By monitoring the growth/inhibitory rate of microbes incubated with various concentrations of AgCu nanoparticles, it was found that at concentrations of 31.3 µg/mL and higher, the growth of *E. coli. S. aureus* and *C. albicans* was inhibited. This matched well with the MIC results for the bacterial cells, whereas the effect dose to inhibit *C. albicans* (15.6 µg/mL) was lower than the MIC (31.3 µg/mL). Lower concentrations were able to reduce the growth of microbes but not completely inhibit their growth, as shown in Figure 1a–c.

### 3.5. Determination of Viable Cells

Fluorescent live–dead assay was used to identify the viability of cells during different time intervals after treatment with AgCu nanoparticles. Bacterial cells were killed at a faster rate than fungal cells, as shown in Figure 2. Error bars denoting the standard error can be found in Appendix A. At 100 µg/mL concentration, less than 99% of *E. coli* and *S. aureus* cells were viable after 5 h of treatment, in comparison to *C. albicans*, with just over 90% cell death. When treated at the MIC, a similar trend was shown but at a slower killing rate. The visual reduction in microbial cells, *C. albicans. E. coli.* and *S. aureus* can be seen in the fluorescent images shown in Figure 3, Figure 4 and Figure 5, respectively.

## 4. Discussion

Antimicrobial resistance is a serious threat to human health, with at least 700,000 global deaths linked to resistant infections annually. The World Health Organisation (WHO) has recently classified the priority list of the top 12 deadliest antibiotic-resistant pathogen strains and urged researchers to find alternative pathways to combat these pathogens [19]. In industry and research, various methods are available to evaluate the antimicrobial activities of different substances; however, most of these methods are only standardised for assessing homogenous compounds in solution (e.g., antibiotics). Currently, there is no standard protocol to accurately measure the antimicrobial efficacy of heterogeneous suspensions (e.g., colloidal silver nanoparticles). Common methods to evaluate the antimicrobial activity of samples include the agar well diffusion method, spectrometry and cell viability dye indicators. However, each method has its disadvantages; the choice of assay and the results obtained are also found to be inconsistent between disciplines [2,20]. In this study, five different designs of in vitro testing methods were conducted to obtain a serial antimicrobial analysis of various metallic-based nanomaterial suspensions against two fungal species and five Gram-negative and three Gram-positive bacteria. These bacterial species were selected from the WHO classified priority list [19]. The fungal species were based on the most common fungal infections with emergent antifungal resistance [21]. The antimicrobial results obtained from each method were compared, and the accuracy, limitations and robustness of each method were evaluated.

The solid agar and resazurin broth methods were initially used to screen the antimicrobial activities of ten nanoparticles at the same concentrations (0.1 *w*/*v*, 1 mg/mL) against the ten chosen pathogens. For nanoparticles that showed antimicrobial potentials (i.e., AgCu, Ag20, Cu10, Cu60 and CuO40) in both screening methods, the broth dilution method was then used to determine the MIC of each type of nanoparticle through the colour change of resazurin. Finally, the highest performance nanoparticle, AgCu, was chosen for further studies using two different quantitative methods: growth curves and live–dead assays. All five individual experiments produced comparable results whilst highlighting the advantages and disadvantages of the testing methods.

### 4.1. Antimicrobial Nanoparticles Screening Approaches

To summarise the antimicrobial screening assays, 10 different nanomaterials were tested against 10 different microbial strains using both the agar well diffusion and the resazurin broth method, where 100 tests were performed in each matrix (excluding triplicates). The agar well diffusion method identified the antimicrobial activity of a suspension through visible inhibitory zones, with a larger diameter size corresponding to higher strength in terms of antimicrobial activity [22]. Table 1 shows that AgCu nanoparticles had the broadest antimicrobial activity as it was effective against all the tested microbes, whilst Cu10 and Cu60 nanoparticles had the highest efficacy toward the Gram-negative *P. aeruginosa*, with a zone of inhibition diameter of 2.35 cm. However, the remaining proportion of the nanoparticle samples did not show antimicrobial activity against other species under the solid agar tests, even though they were reported to have antimicrobial activity [5,10,23]. Thus, this result has provided evidence that direct physical contact is required between the nanoparticles and microbes for the antimicrobial function to be observed. This experiment also reflected the level of the diffusion ability of different nanoparticles through the solid agar; hence, smaller particles are expected to have higher diffusion efficiency than larger particles [2]. Smaller nanoparticles are able to diffuse through agar at a faster rate and have a larger surface area, which is more prone to ion release. It has been reported that physical contact between the nanoparticle and the microbes and that the release of ions are the two main antimicrobial mechanisms for silver [2,24].

Despite the fact that the antimicrobial results obtained through the solid agar method are limited by the particle diffusion rate and the physical and physiochemical properties of individual nanoparticle suspensions, this robust testing method provided not only the semi-quantitative inhibitory zone effect but also provided hints to the mechanistic interaction between each nanomaterial against individual microorganisms. For example, it is interesting to see that all Cu-based nanoparticles had zero zone effect against the majority of the tested microbes, whilst the bimetallic AgCu nanoparticles unexpectedly showed increased antimicrobial efficacies (0.7–1.9 mm) against all species when compared to the standard Ag samples, such observations can be explained by the exhibition of synergistic effect [25].

In contrast, the resazurin broth method allowed full direct contact between microbes and nanoparticles as both are freely suspended in broth; thus, this testing method has resulted in greater antimicrobial responses (Table 2) compared to the solid agar method (Table 1).

The physio-chemical properties of the nanoparticles have been reported to have an effect on antimicrobial properties, which is supported by the screening approaches. The size of nanoparticles is one of the main properties that contribute to their antimicrobial activity, and smaller-sized nanoparticles show higher antimicrobial efficacy [5]. This can be seen in the agar well diffusion method, where smaller-sized silver (Ag20) had more antimicrobial activity compared to larger-sized silver (Ag100). It has been found that the enhanced toxicity of smaller nanoparticles is due to higher surface area, which enables an increased interaction with pathogens and an increased release of ions [2,5,24]. Furthermore, it has been suggested that larger-sized nanoparticles may get trapped as they diffuse through the agar matrix [2,26,27]. Hence, smaller nanoparticles show higher antimicrobial activity due to zones of inhibition as a result of more ion release and the ability to diffuse through the agar. The surface area is also determined by the shape of the nanoparticle. As a result, the physical shape can influence the antimicrobial activity through ion release [28,29]. Cha et al. (2015) found that the shape of nanoparticles contributed to their antimicrobial properties through the compatibility and interaction between nanoparticles and essential microbial enzymes active sites [30].

The zeta potential, the electrostatic surface charge of a nanoparticle, is another physio-chemical property that can influence their antimicrobial activity [31,32]. Microbes commonly have a negative surface charge; thus, nanoparticles with a positive charge will be attracted to microbes and more likely to interact [33]. It has been reported that direct physical contact between nanoparticles and microbes can result in cell death [31].

The agar well diffusion method limits the physical contact between microbe and nanoparticle dispersions; therefore, the antimicrobial activity found using this method is likely to have been a result of ion production, which contributes to the size and shape properties of the nanoparticles. In contrast, the resazurin broth method allows direct contact between nanoparticles and microbes; hence, the antimicrobial activity found using this method is likely to have been a result of interactions between the microbes and nanoparticles through shape compatibility and zeta potential attraction.

In summary, the agar well diffusion method showed a total of 24 positive antimicrobial responses, whereas the broth method was able to detect 75 positive antimicrobial activities in the nanomaterials tested under the same matrix concentrations. All the positive antimicrobial results obtained from the agar well diffusion method were consistent with the results obtained using the resazurin broth methods, with an exception when engineered copper oxide CuO10 was employed. In the case of CuO10, although a clear inhibitory zone of 2.15 cm was observed in the Gram-negative *P. aeruginosa*, live cells were detected using resazurin when the microbes were treated in a broth condition. The clear ring observed in the agar well could be seen as an inhibitory effect in terms of cell production; however, the mother culture appeared to remain metabolically active, leading to resazurin reduction and colour change.

The overall finding is that antimicrobial strength can be indicated through the size of the zone of inhibition produced when using the agar well diffusion method, whilst resazurin dye can only indicate metabolically active cell viability. Therefore, the resazurin method may underestimate the antimicrobial activity of a sample as inhibited, but active cells would cause a colour change. Although the agar well diffusion method is more laborious and may underestimate the antimicrobial effect due to the limited contact between the sample and microbes, nanomaterials that test positive in terms of antimicrobial results using this solid method should be seriously considered as potential candidates. Such positive signs indicate not only the antimicrobial efficacy of the nanomaterials but also the ability to diffuse and release ions under semi-solid conditions, hence their potential versatile applications in biomedical engineering.

### 4.2. Quantitative and Qualitative Evaluations of Antimicrobial Nanoparticles

To determine the effective nanoparticle concentrations that lead to the complete deactivation of each pathogen, the minimum inhibitory concentration (MIC) titre assay was performed. Resazurin indicator dye was again used in this experiment to detect the presence of any viable cells. Other methods to investigate the MIC of a compound include agar dilutions, broth dilutions and gradient method [34]. Due to the solubility and size of nanoparticles in comparison to antibiotics, these methods were unsuitable in terms of testing the antimicrobial activity of nanoparticles; thus, resazurin dye was used to facilitate the interpretation of the results. The utility of resazurin dye also overcomes limit of detection (LoD) and absorbance interference issues, which are often found in nanoparticles or quantum dots with optical profiles. However, resazurin is limited to aerobic microbes since it measures the aerobic respiration of metabolically active microbes [35]. Alternative fluorescent dyes such as XTT (2,3-bis(2-methoxy-4-nitro-5-sulfophenyl)-5-carboxanilide-2H-tetrazolium) can also be used to detect metabolically active cells through the quantification of absorbance; however, these dyes are more expensive, time-consuming and toxic in comparison to resazurin [36,37].

Based on the initial screening results, five nanoparticles which showed highest activities (i.e., AgCu, Ag20, Cu10, Cu60 and CuO40) were chosen for the MIC assay. Microbes were incubated with a gradient of nanoparticle concentrations, and resazurin dye was used to identify the lowest concentration of nanoparticles that did not cause a colour change. The results displayed in Table 3 show that AgCu and Ag20 had the most effective MIC values, as low as 7 µg/mL, whilst copper-based nanoparticles had slightly higher MIC values (31–250 µg/mL). This method has all the advantages of the resazurin broth method and can also be used to compare the concentration efficacy between different samples. The MIC results also corresponded to the zone of inhibition diameter from the agar well diffusion method, where nanoparticles producing larger zones of inhibition required lower concentrations for antimicrobial activity against the respective microbes. Using *E.coli* as an example, four nanoparticles had high MIC values (≥250 µg/mL) results, which were in line with the well diffusion assay, where a small (0.55–0.8 cm) or zero inhibitory zone was observed in the 10 tested nanoparticles (Table 1). In contrast, a larger inhibitory zone was generated by AgCu, which also corresponded to a lower MIC value (7 µg/mL) against *E. coli* obtained using the resazurin MIC assay.

Although Cu nanoparticles were only effective toward *P. aeruginosa* with the agar well diffusion method, it was effective towards 90–100% of the tested microbes with the resazurin broth method. The antimicrobial properties of Cu are suggested to be due to interactions with the cell membrane, which result in changes in cell morphology, cellular penetration and interactions with cellular materials [38,39]. It has also been found that the release of ions contributes to antimicrobial activity [39,40]; however, the agar well diffusion method suggests that the tested Cu nanoparticles in this study have a limited production of ions to result in a zone of inhibition. Cu is an essential trace metal for bacteria and fungi. It is known that microbes can uptake these trace metals; however, at high concentrations, these metals can lead to toxic cellular effect [41,42]. This supports the resazurin results; when the nanoparticles were able to make physical contact with the microbes, they were able to have a higher antimicrobial activity. On the other hand, Ag is not an essential trace metal; hence, the increased antimicrobial activity of bimetallic AgCu may have been a result of two different nanoparticle mechanisms of action leading to a synergistic effect.

Based on the screening methods, AgCu nanoparticles were found to be the best-performing nanoparticles of those tested and were thus selected for more in-depth antimicrobial analysis. In this study, the growth of microbes was monitored overnight in terms of their the optical density measured though the use of spectrometry, as shown in Figure 1a–c. The linear relationship between microbial cells and measured optical density is based on the Lambert–Beer law and can be used to estimate cell count [43,44]. Each data point is presented as the mean absorbance of quadruplicates at each time measurement; data presented with standard deviation as error bars can be found in the Appendix A. As cells multiply, cell density increases, which can be detected by the increase in the turbidity of the sample. Whilst the previous screening methods gave endpoint results after 24 h, this method enables the observation of growth overtime with several measurement points, allowing for the evaluation of growth rate between different samples. The endpoint results obtained from these kinetic experiments were consistent with the MIC values obtained in previous analysis (Table 3). Using *E. coli* as an example (Figure 1b), AgCu nanoparticle concentrations at the MIC (7.8 µg/mL) or higher had no growth after 24 h of incubation, and it was determined in this assay that the microbial growth was inhibited during the first hour of incubation. Hence, when concentrations dropped below the corresponding MIC (3.9 µg/mL to 2 µg/mL), cell growth was observed through increased absorbance at the 10th hour and the 5th hour, respectively. Similar results were observed by Taner et al. [45] when AgCu nanoparticles were tested against *E. coli* at concentrations between 60 and 7.5 µg/mL.

In the case of *C. albicans* (Figure 1a), the curve showed no growth when fungal cells were exposed to nanoparticles at concentrations of 31.3 µg/mL and 15.6 µg/mL despite the MIC result value of 62.5 µg/mL (Table 3). A reason for this is that spectrometry only measures the optical density and does not take into account the viable fungal cells in the mother culture. Thus, the nanoparticles may have inhibited the growth and the cells did not multiply and change the optical density, but the mother cultures were still viable; hence, a colour change was observed when tests were performed using resazurin dye. Another limitation of this method is that suspensions from dead cells and/or nanoparticles may agglomerate at high concentrations and interfere with the reading of the optical density. Limit of detection (LoD) analysis was performed to validate the threshold of the AgCu nanoparticle concentrations, as can be seen in Appendix A. AgCu nanoparticles showed agglomerations, affected absorbance, and generated a spike when the concentrations increased to 500 µg/mL after 20 h of incubation. Pan et al. reported that spectrometry measurement was the least reliable method out of the methods they tested for bacterial cell quantification in the presence of metal oxide nanoparticles [46]. However, the concentrations of nanoparticles they tested using spectrometry were much higher (0.5 mg/mL and 1 mg/mL) than the concentration tested in this paper (62.5 µg/mL to 0.5 µg/mL). Moreover, as the size of fungal cells is significantly larger than bacterial cells (5 µm vs. 1 µm in one dimension), a similar agglomeration effect was also observed when the positive control growth of *C. albican* density increased over time; hence, Appendix A shows an increase in the standard deviation values as the concentrations of cells increased.

In general, the majority of the growth trends showed higher inhibitory efficacies when microbes were exposed to higher concentrations of AgCu nanoparticles, with the exception of when the minimal dose (0.5 µg/mL) was used, the growth of *E. coli* and *S. aureus* appeared to be higher than the positive control (Figure 1b,c). It is known that the debris of dead cells, filamentous growth and leakage of fluorescent proteins may affect the absorbance and the estimation of sample cell density [47,48].

To compensate for the limitations of the above method, fluorescent dyes were used as a supplementary approach to further support the quantifications of cell viability. Herein, propidium iodide and Syto9 dyes were used to identify both dead and live cells. Syto9 has the ability to permeate through live and dead cells and emit green fluorescence when bound to DNA and RNA. On the other hand, propidium iodide can only permeate into dead cells or cells with damaged cell membranes and emits a red fluorescence when bound. Propidium iodide has a higher affinity than Syto9; thus, when they are used together, propidium iodide can displace Syto9 and be used to identify dead and damaged cells [49,50]. A limitation of this method is when observing high numbers of cells unless using a flow cytometer, counting the abundance of cells can become difficult. Furthermore, dead cells that are not intact become undetectable and intermediate states in bacterial samples have also been observed where the fluorescent colour is in relation to the level of cell membrane damage [50,51]. In general, the cost of the dyes is relatively higher than the consumables in the previous methods mentioned in this paper.

In this study, the micrographs obtained from the live/dead assay were further processed using ImageJ v1.53 (National Institution of Health, Bethesda, MD, USA) software to semi-quantify the cell viability (Figure 2). At hourly intervals, the cell viability was investigated for the MIC specific to each microbe and at 100 µg/mL. A concentration higher than all the MIC values was chosen to evaluate the antimicrobial rate of AgCu nanoparticles in relation to the concentration. Overall, the results show that both Gram-positive and negative bacteria were more susceptible to the AgCu nanoparticles when compared to the fungal species *C. albicans*, which is also consistent with the results obtained via the solid agar diffusion method (Table 1), spectrometer growth curves (Figure 1) and the MIC assay (Table 3). The nanoparticles were able to kill a higher percentage of bacteria and acted at a faster rate (0–1 h) than against *C. albicans.* For example, after 5 h of AgCu nanoparticle treatment at the MIC concentration of 7 µg/mL, only 0.12% of *E. coli* was viable in comparison to just over 17.37% of *C. albicans.* Interestingly, yellow and orange cells were observed in the *C. albicans* samples (Figure 3) but not in the bacterial (Figure 4 and Figure 5) samples. This indicated that some of the cell membrane was partially damaged, which led to the permeation of propidium iodide into the cells. The inadequate amount of internalised propidium iodide present had, therefore, resulted in the observation of orange/yellow cells under fluorescent microscopy. This further supports that a longer time is required to damage fungi cells in comparison to bacterial cells. Recent papers are focused on the synthesis and characterisation of bimetallic and alloy AgCu, with only a few that test antimicrobial efficacy. Paszkiewicz et al. (2016) synthesised AgCu nanoparticles and found that they have greater antibacterial effects against *E. coli* and *S. aureus* than anti-fungicidal effect (*C. albicans*) when testing cotton fabric modified with AgCu nanoparticles [52]. Although there are standardised methods to test antimicrobial activity and efficacy on textile products [53], there are currently no standardised methods for testing the antimicrobial properties of nanoparticles.

Whilst the mechanism of action of metallic nanoparticles remains unclear, it is believed that nanoparticles interact and penetrate the cell wall, leading to antimicrobial activity. Bacterial cell walls consist of lipopolysaccharides and peptidoglycan in contrast to fungal yeast cell walls, which consist of an outer cell wall (mannan and cell wall proteins) and an inner wall (chitin, β, 1–6 glucan, β, 1–3 glucan). The difference in susceptibility between the microorganisms could be due to the difference in cell wall composition and structure; hence, AgCu nanoparticles were more effective against bacterial cells than fungal cells [54,55,56]. It is worth noting that bimetallic AgCu nanoparticles also demonstrated strong antiviral potency against both DNA and RNA viruses (>89% viral reduction) [15].

A summary of the method designs used to investigate the antimicrobial properties of metallic-based nanoparticles in this study is shown below (Table 4).

## 5. Conclusions

Comparison studies of five different antimicrobial testing methods were performed, and each method was shown to have advantages and limitations. In general, the choice of selecting a method can be tailored to the sample type and degree of results. The resazurin broth method is good for screening any samples, including those that require direct contact between the sample and microbes; however, the results can be biased and overestimated. Additionally, it cannot detect the strength of the sample or differentiate between inhibition and bactericidal effects. In contrast, the agar well diffusion method was performed under a semi-solid condition with some restriction to physical contact between the sample and microbes. This method indeed was shown to be the most useful; not only was it able to provide information on antimicrobial efficacy through the size of the inhibitory zones but it was also able to identify nanomaterials that produce antimicrobial ions and synergistic effects under a neutral agar environment. The precision of antimicrobial evaluations using the agar well diffusion method should also be highlighted, as all the nanoparticles that showed positive results from this method matched well with the antimicrobial results found in the other four methods used in this study. Therefore, it is important to consider utilising the agar well diffusion method for the selection process of antimicrobial nanoparticles. Nanomaterials show positive antimicrobial activities from the agar well diffusion method, which implies a broader range of biomedical applications (e.g., coatings for medical device and consumables) as the method is not limited to nanoparticles that only exhibit antimicrobial properties through direct contact with the microbes but also through ion diffusion effects.

Using resazurin to determine the MIC of samples is useful in ascertaining the effective concentration of a sample for subsequent work. The spectrometry method is convenient for determining the growth and reduction rates of a sample over time; however, this method cannot detect the cell viability, and both nanoparticles (especially at high levels) and cell debris can interfere with the optical density; hence, limit of detection (LoD) must be performed alongside. Lastly, Syto9 and propidium iodide cell viability dyes were good for semi-quantifying viable cells within the sample; capturing results at regular time intervals also enables the calculation of the microbial reduction of samples over time; however, the sample preparation process in this method was more laborious and only work well at a narrow range of microbial concentrations (*ca*. 3.0–4.0 × 10^4^ CFU/mL).

This paper is of broad multidisciplinary interest as it illustrated the utilities of biological testing methods for advanced materials, along with the limitations of their detections. We believe this manuscript would contribute the necessary technical knowledge to global health services and scientists, especially medical biologists and analytical chemists who are undertaking antimicrobial research, developing and testing devices/accessories with antimicrobial functions.

## Figures and Tables

**Figure 1 bioengineering-10-01259-f001:**
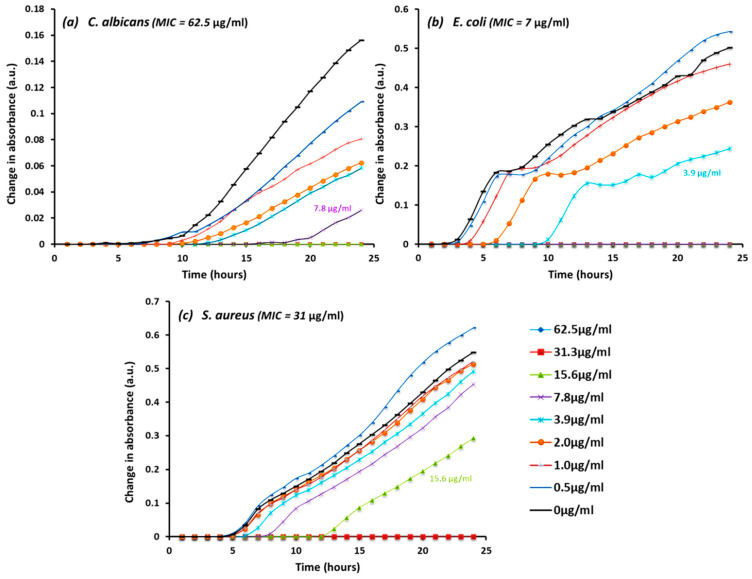
(**a**–**c**): 24 h kinetic growth curve of microbes after exposing (**a**) *C. albicans,* (**b**) *E. coli*, and (**c**) *S. aureus* with AgCu nanoparticles at serial concentrations between 62.5 µg/mL and 0.5 µg/mL. Each data point represents the mean value of quadruplicate time measurements at OD_600_. The corresponding standard deviations can be found in Appendix A.

**Figure 2 bioengineering-10-01259-f002:**
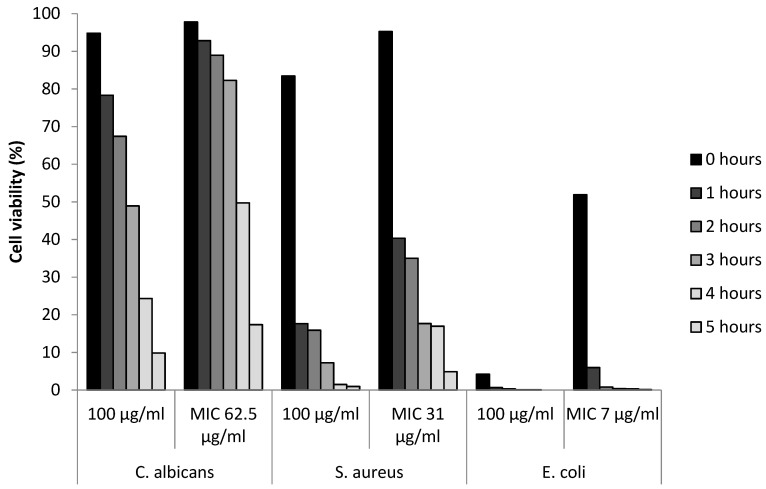
Percentage cell viability of microbes after exposures of different concentrations of AgCu nanoparticles (100 µg/mL and the corresponding MIC) over period of five hours. Results represent three areas of slide count of two replicates.

**Figure 3 bioengineering-10-01259-f003:**
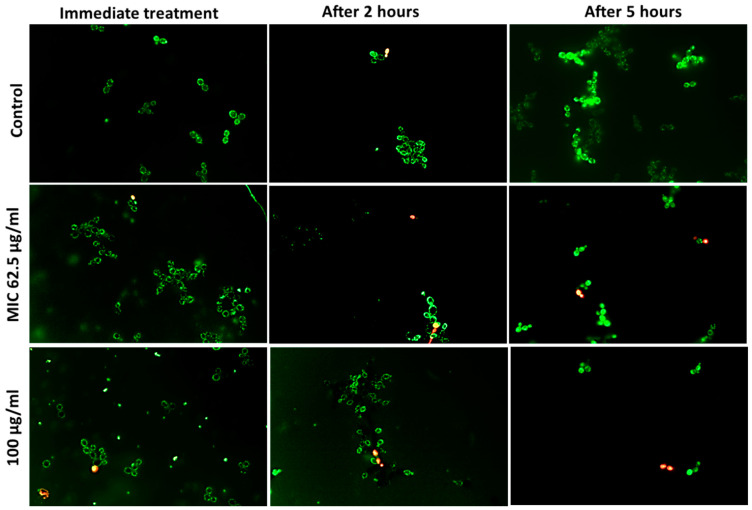
Fluorescent observation of *C. albicans* cell viability after AgCu nanoparticles exposure over five hours at MIC 62.5 µg/mL and concentration of 100 µg/mL. Green fluorescence represents live viable cells and red fluorescence represents dead cells. Intermediate colours yellow and orange indicated partially damaged cells.

**Figure 4 bioengineering-10-01259-f004:**
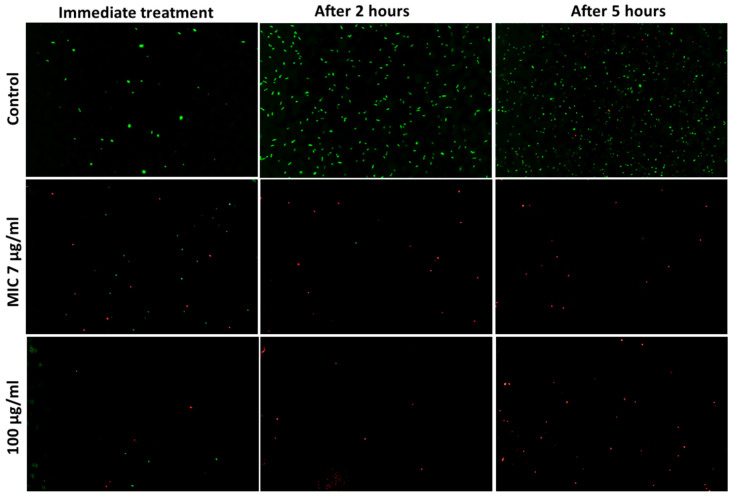
Fluorescent observation of *E. coli* cell viability after exposure of AgCu nanoparticles over five hours at MIC 7 µg/mL and concentration of 100 µg/mL. Green fluorescence represents live viable cells and red fluorescence represents dead cells.

**Figure 5 bioengineering-10-01259-f005:**
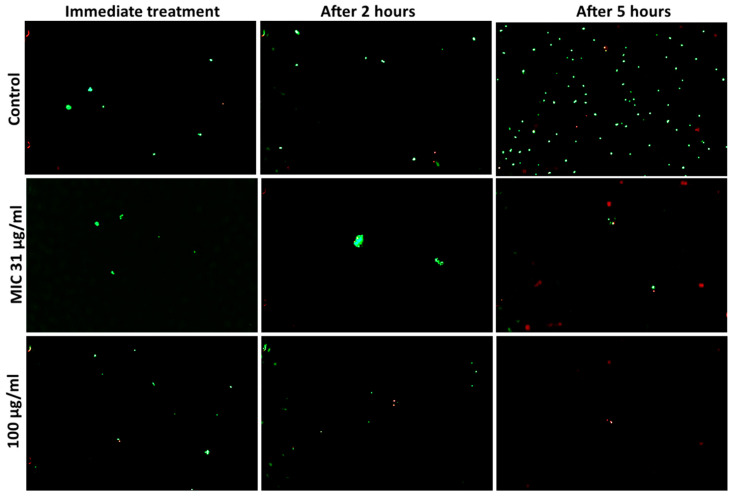
Fluorescent observation of *S. aureus* cell viability after exposure of AgCu nanoparticles over five hours at MIC 31 µg/mL and concentration of 100 µg/mL. Green fluorescence represents live viable cells and red fluorescence represents dead cells.

**Table 1 bioengineering-10-01259-t001:** Zone of inhibition using agar well diffusion method. Results are displayed as mean diameters (measured in cm) of three replicates; ‘0′ zero represents no inhibitory zone.

	Microbe	Ag20	Ag100	AgCu	Cu10	Cu60	Cu90	CuO10	CuO40	CuZn	ZnO
Fungi	*C. albicans*	0	0.85	1	0	0	0	0	0	0	0
	*C. tropicalis*	0	0.65	0.7	0	0	0	0	0	0	0
Gram negative	*A. baumanii*	0.9	0	1.85	0	0	0	0	0	0	0
	*P. aeruginosa*	0	0	1.2	2.35	2.35	2.7	2.15	1.95	0	0
	*K. pneumonia*	0.6	0	1.4	0	0	0	0	0	0	0
	*E. coli*	0.55	0	0.8	0	0	0	0	0	0	0
	*S. typhimurium*	0.6	0	1.4	0	0	0	0	0	0	0
Gram positive	*E. faecium*	0	0	1.4	0	0	0	0	0	0	0
	*S. aureus*	0	0	1.7	0	0	0	0	0	0	0
	*S. pyogenes*	0.8	0	1.9	0	0	0	0	0	1.4	1.4

**Table 2 bioengineering-10-01259-t002:** Broth inhibition assay. ‘+’ represents complete antimicrobial inhibition (no microbial growth), ‘−’ represents absence of antimicrobial activity (microbial growth). Results were obtained from three replications.

Inhibition % Per Strain		Microbe	Ag20	Ag100	AgCu	Cu10	Cu60	Cu90	CuO10	CuO40	CuZn	ZnO
Fungi	60	*C. albicans*	−	+	+	+	+	+	+	−	−	−
	40	*C. tropicalis*	−	+	+	+	−	−	+	−	−	−
Gram negative	80	*A. baumannii*	+	+	+	+	+	+	−	+	+	−
	80	*P. aeruginosa*	+	+	+	+	+	+	−	+	+	−
	90	*K. pneumonia*	+	+	+	+	+	+	−	+	+	+
	90	*E. coli*	+	+	+	+	+	+	−	+	+	+
	80	*S. typhimurium*	+	+	+	+	+	+	−	+	+	−
Gram positive	60	*E. faecium*	+	−	+	+	+	+	+	−	−	−
	80	*S. aureus*	+	+	+	+	+	+	−	+	+	−
	90	*S. pyogenes*	+	+	+	+	+	+	+	−	+	+
		% of microbes susceptible	80	90	100	100	90	90	40	60	70	30

**Table 3 bioengineering-10-01259-t003:** Minimal inhibitory concentrations (MIC in µg/mL) of selected metallic nanoparticles against a range of microbes. The lowest concentration without resazurin colour change after 24 h of incubation with nanoparticle treatment was regarded as the MIC. Results were obtained from three replicates.

	Ag20	AgCu	Cu10	Cu60	CuO40
*C. albicans*	31	62.5	250	250	250
*C. tropicalis*	31	31	125	125	125
*A. baumannii*	15	31	31	31	31
*P. aeruginosa*	7	7	250	250	250
*K. pneumonia*	15	15	250	250	250
*E. coli*	250	7	250	250	>250
*S. aureus*	15	31	125	125	125

**Table 4 bioengineering-10-01259-t004:** Summary of methods used to investigate the antimicrobial properties of metallic-based nanoparticles in this study. ‘X’ indicates that the resources required for the method in that row.

Method	Resources							Time (hours)	Antimicrobial Validation
	Agar Plate	96 Well Plate	Broth	Resazurin	PI & SYTO9	Spectrophotometer	Fluorescent Microscope		
Agar well diffusion	X		X					24	Visible inhibitory zone (qualitative)
Resazurin broth assay		X	X	X				48	Positive/negative antimicrobial activity (qualitative)
Resazurin microtitre MIC assay		X	X	X				48	Minimum effective concentration of tested reagent (quantitative)
Spectrometer growth rate		X	X			X		24	Monitoring of kinetic growth (semi-quantitative)
Live/Dead assay		X	X		X		X	6	Cell viability (qualitative and semi-quantitative)

## Data Availability

Not applicable.

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
