# Peer review of "Applied Methods to Assess the Antimicrobial Activity of Metallic-Based Nanoparticles"

_bioengineering, 2023, doi:10.3390/bioengineering10111259_

Round 1

Reviewer 1 Report (Previous Reviewer 2)

Comments and Suggestions for Authors

After revision the paper can be accepted for publication

Reviewer 2 Report (Previous Reviewer 3)

Comments and Suggestions for Authors

I am satisfied with the revision and can be published in present form.

This manuscript is a resubmission of an earlier submission. The following is a list of the peer review reports and author responses from that submission.

Round 1

Reviewer 1 Report

Comments and Suggestions for Authors

Abstract need to be modified, in order to clarify the main results of the paper.

Section1:

There are a lot of editing errors, please check and modify

I suggest to introduce the chemico-physical properties of the nanoparticles selected for the study. Inparticular in order to correlate to antimicrobial activity can be important analyze and correlate with the biological result the dimensions, the shape, the chemical surface properties, etc.

Figure 2: I suggest to introduce the standard error, in order to well compare the values.

Reviewer 2 Report

Comments and Suggestions for Authors

The paper reports about  the antimicrobial activity of 10 different (by composition and size) metallic based nanoparticles. The work is interesting and can be accepted for publication after the following points are addressed.

1) The work is focused on the biological testing methods. However, a characterization of the nanomaterials used in the experiments would be useful.

2) The author should discuss the effect of copper on the antimicrobial activity. 

3) Was the effect of nanoparticle size detected in the experiment? 

4) Page2, line 88, "dispersions" instead of "solutions"

Reviewer 3 Report

Comments and Suggestions for Authors

Subject: Review of Manuscript Submission Bioengineering-2617199

I have carefully reviewed the manuscript titled "A comparison of applied methods to assess the antimicrobial activity of metallic-based nanoparticles" submitted to Bioengineering for consideration. This paper reports a serial analysis obtained from testing the antimicrobial activities of 10 metallic-based nanomaterials against 10 different pathogens using 5 different in vitro assays, where the technique, limitation, and robustness of each method were evaluated. And I think this study is meaningful and can provide both a basis and directions for future research. It can be accepted after a minor revision.

1.       The title in the original draft should be modified. “A comparison”.

2.       How to realize the mean diameters of three replicates? Briefly, Table 1 illustrates the Ag and Cu with varied diameters. How to measure the size of these materials?

3.       It is recommended that the author summarize the techniques, limitations, and robustness of each method listed in a table, which is not only a summary of the article, but also convenient for readers to read.